# The Co-Occurrence of Post-Traumatic Stress Disorder and Depression in Individuals with and without Traumatic Brain Injury: A Comprehensive Investigation

**DOI:** 10.3390/medicina59081467

**Published:** 2023-08-16

**Authors:** Maja Simonovic, Bojan Nedovic, Misa Radisavljevic, Nikola Stojanovic

**Affiliations:** 1Faculty of Medicine, University of Nis, Nis 18000, Serbia; nedovicbn@gmail.com (B.N.);; 2Center for Mental Health, University Clinical Center, Nis 18000, Serbia; 3Clinic for Neurosurgery, University Clinical Center, Nis 18000, Serbia

**Keywords:** traumatic brain injury, PTSD, depression

## Abstract

Post-traumatic stress disorder (PTSD) is a prevalent psychiatric disorder that often occurs following war trauma. Despite its high prevalence, there is still a lack of comprehensive understanding regarding the mechanisms underlying its progression and treatment resistance. Recent research has shed light on the biological basis of PTSD, with neuroimaging studies revealing altered brain connectivity patterns in affected individuals. In war contexts, traumatic brain injury (TBI) is a common occurrence and is associated with a high prevalence of PTSD. This study aimed to compare the severity of PTSD and depression in patients with and without a history of TBI to shed light on the impact of comorbid TBI on the presentation of PTSD symptoms. To achieve this goal, a cross-sectional study was conducted involving a sample of 60 outpatients who were diagnosed with both PTSD and Depressive Disorder. The inclusion criteria required participants to meet the diagnostic criteria for both disorders using validated tools. The severities of PTSD and depressive symptoms were assessed using scales that have been widely used and validated in previous research. By utilizing these standardized assessment tools, this study aimed to ensure the reliability and validity of the obtained data. The results of this study revealed that patients with comorbid PTSD and TBI exhibited a significantly higher severity of PTSD symptoms compared to those with PTSD only. Specifically, the comorbid group demonstrated higher ratings of symptom intensity across all symptom clusters. These findings are consistent with previous research that has highlighted the impact of comorbid TBI on the intensity and persistence of PTSD symptoms. When controlling for PTSD severity, no significant differences were observed in the severity of depressive symptoms between the two groups. This suggests that the increased depressive symptoms observed in the comorbid group may be primarily driven by the presence of more intense PTSD symptoms rather than TBI per se. The findings highlight the need for an accurate diagnosis of TBI in individuals with PTSD to guide appropriate treatment interventions. Further research is warranted to delve into the underlying mechanisms that contribute to the interaction between TBI and PTSD and to develop targeted interventions for individuals with comorbid PTSD and TBI.

## 1. Introduction

Post-traumatic stress disorder (PTSD) is widely recognized as the most prevalent psychiatric disorder that emerges following exposure to war trauma [1]. While significant progress has been made in understanding the etiology and treatment of PTSD, there remain subsets of patients who exhibit a progressive deterioration of mental status and treatment resistance, which are inadequately explained by current knowledge [2]. Exploring the underlying mechanisms responsible for these observations is crucial for improving our understanding of PTSD and developing more effective interventions.

Extensive research has suggested a biological basis for PTSD, with recent neuroimaging studies providing valuable insights into the altered structural and dynamic functional connectivity associated with this disorder [3,4,5]. These findings have shed light on the neural circuits involved in fear processing, emotional regulation, and memory consolidation, which are disrupted in individuals with PTSD. Importantly, the enduring nature of PTSD symptoms, which can persist for over a decade, exerts a profound impact on the affected individuals, their families, and society as a whole [6].

In the context of war conflicts, traumatic brain injury (TBI) has emerged as a significant concern and is often referred to as a “signature injury” [7]. Blast exposure, which is common in combat settings, is the leading cause of TBI [8]. TBI is characterized by temporary or permanent neurological dysfunction resulting from a head injury, such as loss of consciousness or altered mental status [9]. Among TBI diagnoses, mild TBI is the most prevalent, accounting for approximately 77 percent of cases. It is characterized by confusion or disorientation lasting less than 24 h, loss of consciousness for up to 30 min, memory loss for less than 24 h, and normal brain imaging results.

Research has demonstrated a strong association between TBI and PTSD, with up to half of the individuals with mild TBI meeting the diagnostic criteria for PTSD [10,11]. Furthermore, comorbid depression is common, affecting over one-third of individuals with TBI. This comorbidity of TBI, PTSD, and depression further increases the risk of adverse outcomes, such as suicidal ideation, suicide attempts, and completed suicide [12]. The consequences of TBI extend beyond mental health, as cognitive impairment, alcohol misuse, pain disorders, and unemployment are prevalent among individuals who have experienced TBI as a result of war trauma [13,14,15,16]. Similar associations have been observed in civilian populations, with studies highlighting the links between TBI and depression, as well as behavioral disturbances such as disinhibition, poor decision making, and apathy [17]. Neuropathology in TBI involves axonal injury, which disrupts neural communication and compromises the integrity of networks connecting brain structures to function [18,19]. These disruptions hinder the ability of neural networks to recover, adapt, or re-establish, limiting their involvement in fundamental aspects of self-processing, cognition, and emotional functioning [4]. The interplay between TBI and PTSD is complex and multifaceted, involving neurobiological, psychological, and social factors.

While numerous studies have investigated the association between TBI and PTSD, the majority of research has focused on short-term follow ups. However, clinical experience and the dynamic nature of traumatic axonal injury suggest that the association between TBI and PTSD may have significant implications during long-term follow up [20]. Therefore, there is a need for studies that examine the long-term outcomes and trajectories of individuals with comorbid TBI and PTSD, providing insights into the interplay between these conditions over time.

With this perspective in mind, our study aims to compare the severity of post-traumatic stress disorder and depression in patients with and without a history of TBI. By examining these comorbid conditions and their interplay, we hope to contribute to a deeper understanding of the complex relationship between TBI and PTSD. This research will provide valuable insights into the long-term consequences of these conditions and inform the development of more targeted and effective interventions to improve the outcomes and well-being of individuals affected by comorbid TBI and PTSD.

## 2. Materials and Methods

This cross-sectional study was conducted at the Clinic for Mental Health, Clinical Center Nis, between May 2021 and November 2022. The inclusion criteria required participants to meet diagnostic criteria for both post-traumatic stress disorder (PTSD) and Depressive Disorder using The Structured Clinical Interview for DSM-IV Axis I Disorders (SCID-I, modified) with or without comorbid TBI [21,22]. Exclusion criteria included age over 65 years, a lifetime history of psychotic disorder, severe somatic illness, alcohol dependence disorder, and epilepsy. The sociodemographic characteristics of the participants were assessed using a semi-structured clinical interview.

Assessment of Head Trauma and Traumatic Brain Injury: The history of head trauma was determined based on previous clinical diagnoses of TBI. Patients were screened positive for TBI if they had been clinically diagnosed with head injuries resulting from blast or explosion, vehicular accidents, fragment or bullet wounds above the shoulder, falls, objects hitting the head, or if they had been hit by another person, with severity measured in terms of the patient’s level of consciousness and neurological functioning, duration of loss of consciousness, or post-traumatic amnesia [23]. The medical records of the participants were carefully reviewed to collect information regarding the type and severity of the head trauma they had experienced.

Assessment of PTSD and Depressive Symptoms: The severity of PTSD symptoms was assessed using the Clinician-Administered PTSD Scale for DSM-IV (CAPS) [24]. The CAPS is a clinician-administered interview that evaluates the presence and severity of PTSD symptoms based on the 17 DSM-IV criteria. The frequency and intensity of each symptom were evaluated on a zero to four Likert scale. The CAPS yields scores for three subscales (Criterion B re-experiencing, Criterion C avoidance and numbing, and Criterion D hyperarousal), as well as overall frequency, intensity, and total PTSD scores. The CAPS was administered to patients by licensed senior psychiatrists who had undergone rigorous training in CAPS administration. To ensure the reliability of the assessments, a random selection of 20 percent of interviews was scored by a blinded clinician for inter-rater reliability (ICC = 0.95). The CAPS is a widely recognized and extensively validated instrument used in research and clinical settings for the diagnosis and measurement of PTSD symptoms.

The overall severity of depressive symptoms over the past month was assessed using the Montgomery–Asberg Depression Scale (MADRS) [25]. The MADRS is a well-established and widely recognized clinician-rated scale specifically designed to assess the severity of depressive symptoms. MADRS consists of 10 items, each targeting distinct aspects of depression. By evaluating a range of symptoms, it provides a more comprehensive assessment of depressive states compared to scales with fewer items. The ratings on the Likert scale (ranging from 0 to 6) used in MADRS offer a graded and standardized approach to quantifying the severity of symptoms. This allows for consistent and reliable comparisons between patients and facilitates the monitoring of changes in symptom severity over time. The total score range of 0 to 60 provides a quantitative representation of the overall depression severity, enabling researchers and clinicians to categorize individuals into different levels of depressive states, from mild to severe. The choice of MADRS in this study may also be influenced by its relevance in the context of comorbid conditions. The mention of a cut-off score of 20 for the diagnosis of depression comorbid with PTSD suggests that we were interested in understanding depression’s specific impacts in the presence of post-traumatic stress disorder (PTSD). Similar to the CAPS, 20 percent of interviews were randomly selected and scored by a blinded clinician to assess inter-rater reliability (ICC = 0.95).

The overall severity of depressive symptoms over the past month was also evaluated using the Hamilton Depression Rating Scale (HDRS) [26]. The HDRS is a widely used clinician-rated scale designed to assess the severity of depression in individuals. It comprises 17 items, each rated on a Likert scale from 0 (not present) to 4 (severe), providing a total score range of 0 to 52. In this study, a cut-off score of 18 was considered significant for the diagnosis of depression comorbid with PTSD, in line with previous research and clinical guidelines. To ensure the reliability of HDRS assessments, a rigorous approach was adopted. Twenty percent of the interviews were randomly selected, and the recordings were scored by a blinded clinician. The inter-rater reliability was calculated using the Intraclass Correlation Coefficient (ICC), which yielded a high ICC value of 0.91.

Statistical Analysis: All statistical analyses were performed using the STATA software package v15.1 (Stata Corporation, College Station, Texas). Descriptive statistics were used to summarize the demographic characteristics of the participants, which were presented as means with standard deviations (SD) for continuous variables or percentages for categorical variables. Group comparisons between the PTSD+TBI and PTSD-only groups were analyzed using appropriate inferential statistical tests, including independent t-tests, MANOVA, or MANCOVA tests. The significance level for all analyses was set at a two-sided *p*-value of 0.05, indicating statistical significance.

Ethical Considerations: This study was conducted following the ethical principles outlined in the Declaration of Helsinki. Ethical approval was obtained from the institutional review board of the Clinic for Mental Clinical Center Nis. All participants provided written informed consent before participating in this study. The confidentiality of participants’ personal information was strictly maintained, and data were anonymized and stored securely. Participation in this study was voluntary, and participants were free to withdraw at any time without consequences. This study posed no physical or psychological harm to the participants, and appropriate measures were taken to ensure their well-being throughout this study.

## 3. Results

A total of 60 patients were included in this study, with 37 patients (61.67 percent) diagnosed with mild traumatic brain injury (TBI) and 23 patients (38.33 percent) without a TBI diagnosis. The demographic characteristics, including age, education, and marital status, did not show significant differences between the two groups (data not shown), indicating that the groups were comparable in terms of these variables.

An independent-samples t-test was conducted to compare the severity of post-traumatic stress disorder (PTSD) symptoms between the group with comorbid PTSD and TBI (PTSD with TBI) and the group with PTSD only. The results revealed significant group differences for the Clinician-Administered PTSD Scale (CAPS) total score (t(59) = 3.21, *p* = 0.001), CAPS total symptom frequency (t(59) = 2.61, *p* = 0.009), and CAPS total symptom intensity (t(59) = 3.71, *p* = 0.001). In each case, scores were significantly higher for the PTSD + TBI group compared to the PTSD-only group Table 1. This suggests that individuals with comorbid TBI and PTSD experienced more severe PTSD symptoms than those with PTSD alone.

To further examine the specific factors contributing to the overall group difference, a one-way between-group multivariate analysis of variance (MANOVA) was conducted on the CAPS criterion subscale scores (Criteria B, C, and D). The MANOVA revealed a significant main effect of TBI on the combined dependent variables (F(3, 56) = 5.28, *p* = 0.002, Wilks’ Lambda = 0.78). The subsequent analysis of the simple effects on subscale and individual item scores provided further insights into the group differences, as shown in Table 2.

Specifically, individuals with comorbid TBI and PTSD exhibited higher scores on re-experiencing, avoidance and numbing, and hyperarousal symptoms compared to those with PTSD alone. Within cluster B, which includes symptoms related to intrusive recollections, distressing dreams, and a sense of reliving the traumatic event, the comorbid group exhibited significantly higher intensity levels. Similarly, cluster C, characterized by symptoms of avoidance, showed significantly higher intensity in individuals with comorbid PTSD and TBI, reflecting more pronounced efforts to avoid trauma-related thoughts and feelings. Moreover, in cluster D, consisting of symptoms of hyperarousal, the PTSD with TBI group demonstrated significantly higher intensity scores, indicating a potentially more severe impact of the trauma on these individuals.

Additionally, independent-sample *t*-tests were performed to examine the potential group differences on the Montgomery–Asberg Depression Rating Scale (MDRS) and Hamilton Depression Rating Scale (HDRS) scores, as shown in Table 3. The results showed that the PTSD with TBI group did not report significantly higher MDRS or HDRS total scores. However, significant differences were observed within the MDRS for reduced sleep (t(59) = 2.88, *p* = 0.004) and within the HDRS for insomnia in the early hours of the morning (t(59) = 2.12, *p* = 0.034) and general somatic symptoms (t(59) = 2.68, *p* = 0.007). These findings suggest that individuals with comorbid TBI and PTSD may experience more sleep disturbances and general somatic symptoms compared to those with PTSD alone.

To determine the contribution of PTSD severity to the aforementioned symptoms (reduced sleep, insomnia in the middle of the night, general somatic symptoms), a one-way between-group multivariate analysis of covariance (MANCOVA) was conducted with the total CAPS score as a covariate. The results showed a significant main effect of the PTSD covariate (F(3, 55) = 14.16, *p* < 0.001, Wilks’ Lambda = 0.55). This indicates that the severity of PTSD significantly influenced the severity of the reported symptoms. However, there was no main effect of the TBI status after controlling for PTSD severity (F(3, 55) = 1.21, *p* = 0.32, Wilks’ Lambda = 0.94). This suggests that the association between TBI and the specific symptoms of reduced sleep, insomnia, and general somatic symptoms can be attributed to the severity of PTSD.

The demographic characteristics of the study population revealed that the sample consisted predominantly of males (93.33 percent), with a mean age of 48.38 years (SD = 7.04, range = 38–64). All participants were Caucasian, with 80 percent of patients being married, 5 percent single, and 15 percent divorced. In terms of education, 11.67 percent of patients completed elementary school only, 85 percent completed secondary school, and 3.33 percent had earned a master’s degree. These demographic characteristics provide a snapshot of the sample and reflect the specific population from which the participants were recruited.

## 4. Discussion

In this study, we aimed to compare the severity of war-related post-traumatic stress disorder (PTSD) and depression in patients with PTSD and traumatic brain injury (TBI) to those with PTSD only. Our findings revealed that the group with comorbid PTSD and TBI (PTSD with TBI) exhibited a higher severity of PTSD symptoms, as measured using the Clinician-Administered PTSD Scale (CAPS) total score. This finding is consistent with recent studies [27,28] and highlights the impact of comorbid TBI on the intensity of PTSD symptoms. In particular, the PTSD with TBI group demonstrated higher ratings of symptom intensity in the re-experiencing, avoidance/numbing, and hyperarousal symptom clusters, as well as higher frequency ratings in the re-experiencing symptom cluster (Table 1).

The increased severity of PTSD symptoms in individuals with comorbid TBI may be attributed to the mechanisms of brain injury [29,30]. Biological models propose that PTSD involves an exaggerated amygdala response and an impaired functioning of the ventral/medial prefrontal cortex and hippocampus, leading to difficulties in fear inhibition, increased attention to trauma-related stimuli, and impaired learning and memory [31]. Similarly, TBI often results in damage to the prefrontal cortex, ventral frontal lobe, and anterior temporal lobe due to shearing forces, which may disrupt brain networks involved in anxiety regulation [29]. Consequently, individuals with a history of TBI may experience more intense PTSD symptoms. The overlap in affected brain regions and disrupted neural networks in both TBI and PTSD may contribute to the heightened symptomatology observed in comorbid cases.

Furthermore, TBI can affect distributed brain networks and result in behavioral disturbances. The increased severity of PTSD symptoms, such as re-experiencing in patients with a history of TBI, may be a consequence of altered structural and functional connectivity of neural networks connecting prefrontal brain structures, including the hippocampus, dorsolateral prefrontal cortex, and orbitofrontal cortex [32]. Additionally, the decreased integrity of axonal connections between these structures and temporal and occipital areas, which are vulnerable to TBI, may contribute to heightened symptomatology [33].

The co-occurrence of TBI and PTSD may induce a shift in brain state from a high-level processing of contextual and mnemonic stimuli mediated by the hippocampus and prefrontal cortex (working memory) to a more primitive response driven by the amygdala, involving time-locked sensory associations and the expression of species-specific defense responses [34].

Executive impairment resulting from head injury has been linked to the perseveration of traumatic events and re-experiencing symptoms [35]. Moreover, studies have highlighted the role of altered long- and short-distance connections and functional connectivity in both PTSD and TBI [36]. Long-distance connections are crucial for dynamic network reconfiguration during cognitive tasks. Brain regions vulnerable to TBI coincide with those contributing to controllability, with transitions to pathological states (e.g., intrusion, avoidance, and hyperarousal symptoms in comorbid PTSD), potentially reflecting reduced control efficiency [37]. Regions with a high number of long-distance paths to areas of high activity are optimal controllers, and their effectiveness can be altered following TBI. Notably, PTSD, particularly without a history of TBI, has been associated with reduced short connections and small-worldness, indicative of a reduced structure and increased randomness. In contrast, TBI is associated with an increase in short connections, small-worldness, and hyperconnectivity of functional networks, which may disrupt network topology by mixing connections across modules [38].

Regarding the measures of depressive psychopathology, the PTSD with TBI group did not report higher scores on the Montgomery–Asberg Depression Rating Scale (MADRS) or the Hamilton Depression Rating Scale (HDRS) total scores. However, significant differences were found within the MADRS for the item “Reduced sleep” and within the HDRS for the items “Insomnia: early in the morning” and “General somatic symptoms”. Previous research suggests that individuals with comorbid PTSD and TBI may experience poorer physical health and persistent sleep disturbances [39]. Nevertheless, when controlling for PTSD severity, the clinically significant group differences disappeared, indicating that these outcomes may be primarily driven by the presence of more intense PTSD symptoms rather than TBI per se, which is consistent with previous studies [40,41].

While our study provides valuable insights into the severity of PTSD symptoms in individuals with and without comorbid TBI, certain limitations should be acknowledged. The cross-sectional design of this study limits our ability to establish causal relationships between TBI and the severity of PTSD symptoms. Longitudinal studies would be beneficial to better understand the temporal dynamics and potential bidirectional influences between TBI and PTSD. The sample size of our study, while sufficient for the analyses conducted, may limit the generalizability of the findings to larger populations. Future studies with larger and more diverse samples would strengthen the robustness and external validity of our results. This study primarily relied on self-reported and clinician-administered assessments to measure PTSD and depressive symptoms. The subjective nature of these measures could introduce response biases or social desirability effects. Incorporating objective measures, such as neuroimaging techniques or physiological markers, could provide a more comprehensive understanding of the underlying neurobiological mechanisms. One of the limitations of our study lies in the screening process for traumatic brain injury (TBI) in the participants. Although we employed a comprehensive assessment battery and clinical interviews to identify individuals with a history of TBI, it is essential to acknowledge that some cases of mild TBI may have been undetected or under-reported. Mild TBIs, also known as concussions, can sometimes go unnoticed or may not be linked to the traumatic event, leading to potential underestimations of TBI prevalence in our sample. Moreover, the reliance on the self-reported history of TBI and medical records may introduce recall biases and inaccuracies in reporting past head injuries. Some participants might not recall or may not have access to medical records that document prior TBIs accurately. Additionally, the lack of objective measures, such as neuroimaging or neuropsychological assessments, to confirm the occurrence and severity of TBIs may limit the accuracy of our TBI classification. Furthermore, this study’s inclusion criteria may have influenced the TBI screening process. Participants were selected based on their PTSD status, which may have introduced a bias toward recruiting individuals with more severe PTSD symptoms and potentially more significant trauma exposure. Consequently, the prevalence and characteristics of TBI in our sample might not be representative of the broader population of individuals with comorbid PTSD and TBI. To address these limitations, future studies could consider employing more comprehensive and objective methods for TBI screening, such as advanced neuroimaging techniques like diffusion tensor imaging (DTI) or neurocognitive assessments to detect subtle signs of brain injury. Additionally, employing larger and more diverse samples, including individuals from various trauma populations, would enhance the generalizability of findings and provide a more comprehensive understanding of the TBI prevalence and its impact on PTSD symptomatology. Lastly, the study population consisted predominantly of male Caucasian participants, which may not fully represent the diversity of individuals with PTSD and TBI. Future research should aim to include a more diverse sample to account for potential ethnic and gender differences in symptomatology and treatment outcomes.

Strengths of this study include the use of diagnostic clinician-rated assessments for PTSD and depression, enhancing the reliability of our findings. Additionally, the examination of data at multiple time points allowed for cross-sectional analysis, further supporting the robustness of our results.

In conclusion, our study demonstrates that individuals with a history of TBI exhibit more severe PTSD symptoms. The higher intensity of symptoms across all symptom clusters in the comorbid group, along with their prolonged persistence, underscores the importance of accurately diagnosing TBI in individuals with PTSD to inform tailored treatment approaches.

The mechanisms underlying the interaction between TBI and PTSD involve altered brain functioning, disrupted neural networks, and changes in connectivity patterns. Future research should further explore these mechanisms to develop targeted interventions for individuals with comorbid PTSD and TBI. The identification of specific neural circuitry and pathways implicated in comorbid PTSD and TBI could guide the development of novel therapeutic approaches, such as brain stimulation techniques or cognitive interventions aimed at restoring network connectivity and improving symptomatology.

Additionally, longitudinal studies investigating the long-term outcomes and trajectories of individuals with comorbid PTSD and TBI are needed to inform early intervention strategies and optimize treatment outcomes. The integration of findings from neuroimaging, clinical assessments, and treatment outcome measures will contribute to a comprehensive understanding of the complex relationship between TBI and PTSD and facilitate the development of personalized interventions for this vulnerable population.

## Figures and Tables

**Table 1 medicina-59-01467-t001:** CAPS scores for PTSD and PTSD with TBI groups.

Variable	PTSD N = 23 Mean (SD)	PTSD with TBI N = 37 Mean (SD)	t	*p*
**Overall** **PTSD** **Symptoms**				
Total Score	85.83 (19.19)	100.68 (13.38)	3.21	0.001
Total Frequency Score	46.22 (10.62)	52.86 (7.39)	2.61	0.009
Total Intensity Score	39.61 (9.25)	47.81 (6.56)	3.71	0.001
**Reexperiencing** **symptoms**				
Criterion B Total Score	26.30 (7.40)	32.49 (5.42)	3.50	0.001
Criterion B Overall Frequency	14.30 (3.92)	17.08 (2.81)	2.65	0.008
Criterion B Overall Intensity	12 (3.80)	15.40 (2.89)	3.68	0.001
**Avoidance and ** **numbing** **symptoms**				
Criterion C Total Score	33.52 (7.86)	37.57 (6.31)	2.09	0.043
Criterion C Overall Frequency	17.69 (4.52)	19.57 (3.69)	1.67	0.102
Criterion C Overall Intensity	15.82 (3.71)	18 (2.91)	2.39	0.022
**Hyperarousal** **symptoms**				
Criterion D Total Score	26 (6.45)	30.62 (4.20)	2.90	0.004
Criterion D Overall Frequency	14.22 (4.04)	16.22 (2.22)	1.95	0.051
Criterion D Overall Intensity	11.78 (2.81)	14.40 (2.36)	3.73	0.001

**Table 2 medicina-59-01467-t002:** CAPS scores for PTSD and PTSD with TBI individual items.

Variable	PTSD N = 23 Mean (SD)	PTSD with TBI N = 37 Mean (SD)	t	*p*
B1F	2.39 (1.08)	3.16 (0.87)	2.84	0.004
B2F	2.96 (0.98)	3.43 (0.80)	1.88	0.060
B3F	2.56 (1.08)	3.19 (0.81)	2.27	0.023
B4F	3.13 (0.87)	3.54 (0.56)	1.78	0.075
B5F	3.26 (0.91)	3.75 (0.43)	2.10	0.036
B1I	2.04 (0.82)	2.70 (0.81)	3.03	0.004
B2I	2.61 (0.89)	3.13 (0.75)	2.42	0.015
B3I	1.91 (0.95)	2.94 (0.88)	3.88	0.000
B4I	2.61 (0.89)	3.16 (0.69)	2.51	0.012
B5I	2.91 (0.99)	3.45 (0.60)	2.13	0.033
C1F	2.26 (0.96)	2.73 (0.93)	1.92	0.055
C2F	2.26 (0.96)	2.65 (0.89)	1.70	0.088
C3F	1.30 (0.63)	1.43 (0.73)	0.78	0.437
C4F	2.91 (0.85)	3.05 (0.81)	0.64	0.528
C5F	2.87 (0.97)	3.40 (0.56)	2.17	0.030
C6F	3 (0.80)	3.27 (0.73)	1.39	0.165
C7F	2.96 (0.93)	3.03 (0.90)	0.405	0.685
C1I	2.04 (0.82)	2.43 (0.73)	1.86	0.070
C2I	2 (0.80)	2.38 (0.64)	2.38	0.017
C3I	1.26 (0.69)	1.43 (0.80)	0.88	0.382
C4I	2.69 (0.70)	2.81 (0.66)	0.63	0.531
C5I	2.52 (0.73)	3.13 (0.53)	3.34	0.001
C6I	2.65 (0.57)	3 (0.62)	2.35	0.019
C7I	2.52 (0.66)	2.81 (0.74)	2.10	0.036
D1F	3.56 (0.84)	3.86 (0.42)	1.61	0.108
D2F	3.43 (0.73)	3.64 (0.48)	0.98	0.327
D3F	2.09 (0.10)	2.67 (0.71)	2.47	0.018
D4F	2.52 (1.27)	3.08 (0.95)	1.72	0.086
D5F	2.61 (1.16)	2.95 (1.05)	1.26	0.208
D1I	3 (0.95)	3.67 (0.58)	3.09	0.002
D2I	2.78 (0.85)	3.24 (0.72)	2.16	0.037
D3I	1.87 (0.69)	2.48 (0.65)	3.43	0.001
D4I	2.04 (0.93)	2.54 (0.90)	2.04	0.047
D5I	2.09 (0.79)	2.46 (0.99)	1.61	0.114

**B1**—Recurrent and intrusive distressing recollections of the event, including images, thoughts, or perceptions. **B2**—Recurrent distressing dreams of the event. **B3**—Acting or feeling as if the traumatic event were recurring (includes a sense of reliving the experience, illusions, hallucinations, and dissociative flashback episodes, including those that occur on awakening or when intoxicated). **B4**—Intense psychological distress at exposure to internal or external cues that symbolize or resemblean aspect of the traumatic event. **B5**—Physiological reactivity on exposure to internal or external cues that symbolize or resemble an aspect of the traumatic event. **C1**—Efforts to avoid thoughts, feelings, or conversations associated with the trauma. **C2**—Efforts to avoid activities, places, or people that arouse recollections of the trauma. **C3**—Inability to recall an important aspect of the trauma. **C4**—Markedly diminished interest or participation in significant activities. **C5**—Feeling of detachment or estrangement from others. **C6**—Restricted range of affect (e.g., unable to have loving feelings). **C7**—Sense of a foreshortened future (e.g., does not expect to have a career, marriage, children, or a normal life span). **D1**—Difficulty falling or staying asleep. **D2**—Irritability or outbursts of anger. **D3**—Difficulty concentrating. **D4**—Hypervigilance. **D5**—Exaggerated startle response. **F**—Frequency. **I**—Intensity.

**Table 3 medicina-59-01467-t003:** Additional Comparisons for PTSD and PTSD with TBI groups.

Variable	PTSD N = 23 Mean (SD)	PTSD with TBI N = 37 Mean (SD)	t	*p*
MADRS				
Total score	31.35 (6.94)	34.30 (5.98)	1.63	0.103
Apparent Sadness	3.56 (1.04)	3.62 (0.95)	0.21	0.834
Reported Sadness	4.00 (0.95)	3.84 (0.99)	0.63	0.530
Inner Tension	4.48 (0.66)	4.81 (0.62)	1.94	0.059
Reduced Sleep	4.70 (0.93)	5.35 (0.68)	2.87	0.004
Reduced Appetite	1.74 (1.71)	2.51 (1.64)	1.60	0.109
Concentration Difficulties	2.65 (0.98)	2.91 (1.01)	1.01	0.317
Lassitude	3.74 (1.18)	3.81 (1.02)	0.24	0.811
Inability to Fee	3.96 (0.82)	4.08 (0.83)	0.57	0.573
Pessimistic Thoughts	1.61 (0.84)	2.16 (1.12)	1.74	0.082
Suicidal Thoughts	0.91 (1.12)	1.19 (1.08)	1.11	0.267
HDRS				
Total score	26.69 (6.64)	29.65 (4.81)	1.85	0.072
Depressed mood	2.48 (0.90)	2.46 (0.80)	0.08	0.935
Feelings of guilt	1.22 (0.90)	1.38 (0.59)	0.76	0.453
Suicide	0.87 (1.10)	0.68 (0.75)	0.35	0.729
Insomnia: early in the night	1.83 (0.39)	1.97 (0.50)	1.60	0.110
Insomnia: middle of the night	1.87 (0.34)	2.02 (0.37)	1.71	0.087
Insomnia: early hours of the morning	1.78 (0.52)	2.02 (0.37)	2.12	0.034
Work and activities	2.48 (0.99)	2.70 (0.94)	0.87	0.390
Retardation	1.30 (0.97)	1.38 (1.04)	0.28	0.780
Agitation	1.65 (0.93)	2.11 (0.81)	1.93	0.060
Anxiety psychic	2.26 (0.62)	2.35 (0.71)	0.52	0.607
Anxiety somatic	2.70 (0.63)	2.94 (0.47)	1.63	0.110
Somatic symptoms gastro-intestinal	1.26 (0.75)	1.38 (0.76)	0.59	0.560
General somatic symptoms	1.61 (0.58)	1.95 (0.32)	2.68	0.007
Genital symptoms	1.48 (0.66)	1.62 (0.64)	0.99	0.324
Hypochondriasis	1.22 (0.73)	1.51 (0.77)	1.49	0.142
Loss of weight	0.69 (0.70)	1.08 (0.76)	1.91	0.056
Insight	0 (0)	0.08 (0.36)	1.12	0.261

## Data Availability

The data presented in this study are available upon request from the corresponding author. The data are not publicly available due to privacy restrictions.

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
