# Peer review of "The Co-Occurrence of Post-Traumatic Stress Disorder and Depression in Individuals with and without Traumatic Brain Injury: A Comprehensive Investigation"

_medicina, 2023, doi:10.3390/medicina59081467_

Round 1

Reviewer 1 Report

The study by Dr. Simonovic and colleagues is a cross-sectional study of the patients with PTSD and PTSD with TBI. Authors have conducted the study over a period of about 2 years. The methods of the study with the use of the appropriate techniques for the analysis makes the study well enough to read and comprehend. The study is quite well written with appropriate tables.

I have a few minor suggestions which once corrected can be useful for the publication.

1. Would encourage the authors to use paragraphs for the write-up with appropriate headings for better readability.

2. Limitations and conclusions for the study should be presented clearly.

3. Please clarify if the administered scales like CAPS is considered a standard practice or it was for the purpose of the study since there is no funding source mentioned.

Author Response

Dear Reviewer,

Thank you for taking the time to review our study. We appreciate your positive feedback and are grateful for your valuable suggestions to enhance the manuscript. We have carefully considered each of your points, and we made the necessary revisions to improve the overall clarity and readability of the paper.

  • Paragraphs and Headings: We agree with your suggestion to use paragraphs and appropriate headings in the write-up. We reorganized the content into well-structured paragraphs, and we added clear headings to sections that warrant them. This will enhance the readability and flow of the manuscript.
  • Limitations and Conclusions: We acknowledge the importance of clearly presenting the limitations and conclusions of our study. In the revised manuscript, we provided a dedicated section for discussing the limitations of our research, addressing potential biases, and outlining areas for future investigations. Additionally, we revised and elaborated on our conclusions to ensure they are well-explained and supported by the study findings.
  • Use of CAPS Scale: We apologize for the oversight in not providing sufficient information about the use of the Clinician-Administered PTSD Scale (CAPS). The CAPS is a widely accepted and standardized tool for assessing PTSD severity and is commonly used in research and clinical practice. In our study, we selected CAPS due to its well-established validity and reliability in diagnosing and measuring PTSD symptoms. However, we will make sure to clarify this point in the revised manuscript, explicitly mentioning that CAPS was chosen as a standard practice for assessing PTSD in our patient population.

Once again, we appreciate your thoughtful comments and suggestions, which will undoubtedly strengthen our manuscript. We made the necessary changes as per your recommendations and resubmitted the revised version.

Thank you for your time and consideration.

Sincerely,

Dr. Bojan Nedovic

Reviewer 2 Report

The authors provided a detailed study comparing PTSD and PTSD + TBI symptoms. This is a useful and interesting study, valuable to evaluation of patients particularly after war traumas.

Here are several minor revision that the authors should consider:

1. Abstract should be more concise

2. Introduction should be divided into several paragraphs making it easier to read

3. Methods should be divided into sections

4. Results: Table 2 variables not well explained in the Table itself and their effects not described fully in the text.

5. Discussion: paragraphing is again needed.

OK

Author Response

Dear Reviewer,

Thank you for reviewing our study. We appreciate your positive feedback and find your suggestions to be valuable in improving the clarity and readability of our manuscript. We have carefully considered each of your points and made the necessary revisions to address these concerns.

  • Abstract Conciseness: We agree that the abstract should be more concise. In the revised version, we condensed the abstract while ensuring that all essential information is retained. We aimed to provide a clear and succinct summary of the study's objectives, methods, key findings, and conclusions.
  • Introduction Organization: We acknowledge the need to improve the organization of the introduction. To enhance readability, divided the introduction into several paragraphs, each focused on specific aspects. This will help to present the information more coherently.
  • Methods Division: We appreciate your suggestion to divide the methods section into sections. In the revised manuscript, we separated the methods into distinct sections. This will provide a clearer structure and make it easier for readers to navigate through the methodology.
  • Results and Table 2: We apologize for any confusion caused by the presentation of Table 2 and the variable descriptions in the text. In the revised version, we provided a more detailed explanation of the variables included in Table 2. Additionally, we ensured that the effects of these variables are thoroughly described in the text, allowing readers to understand the significance of the results.
  • Discussion Paragraphing: We agree that paragraphing in the discussion section needs improvement. In the revised manuscript, we organized the discussion into well-defined paragraphs, each addressing specific aspects of the study's findings, their implications, and the potential mechanisms underlying the observed results.

Once again, we thank you for your thoughtful review and valuable feedback. We implemented the suggested revisions to enhance the manuscript's clarity and cohesiveness. Your insights will undoubtedly contribute to the overall quality of our research.

Sincerely,

Dr. Bojan Nedovic
